# ACHIEVING DYNAMIC ACCURACY IN MACHINE-LEARNED CG POTENTIALS BY MODULATING POTENTIAL ENERGY LANDSCAPE

## ABSTRACT

In this paper, we introduce a coarse-grained (CG) model designed to reproduce the structure and dynamics of all-atom systems. Our approach combines a graph neural network potential and a high-frequency potential energy surface landscape function to effectively capture essential features of the fine-grained atomistic model. The Neural-Network potential accurately captures complex atomic interactions using learned representations and can be effectively parameterized to reproduce distribution functions from high-fidelity all-atom (AA) simulations. Nevertheless, such parameterization inherently smoothens out the AA Energy landscape, resulting in the loss of information required for capturing the system dynamics. We, therefore, provide a route to enrich the ML CG potentials for bulk systems by emulating the AA landscape within the mapped CG ensemble by augmenting the GNN potential with a high-frequency potential term, thereby providing an accurate representation of CG dynamics as well as the structure. We demonstrate the utility of our framework by reproducing the Radial Distribution Function (RDF) and the mean-squared displacement (MSD) of various AA and CG systems. Notably, we apply our methodology to coarse-grain the widely used SPC/E water model, thereby providing compelling evidence of the fidelity of our model to coarse-grain complex systems, which include electrostatic and multibody effects. Our work takes a significant step towards more efficient and accurate simulations of complex systems using coarse-grained methodologies.

## 1 INTRODUCTION

Molecular dynamics (MD) simulations have been a crucial tool for investigating a wide range of physical, chemical, and biological systems at the molecular level (Hospital et al. (2015)). For instance, MD simulations have been used to study important biomolecular systems and processes like protein folding, enzyme catalysis, ligand binding, and allosteric regulation. (Freddolino et al. (2010); Faucher et al. (2019); Sarikaya et al. (2003)). To enhance computational efficiency, CG simulations have been proposed (Levitt & Warshel (1975)). Some of the advanced CG approaches particularly aim to bridge larger spatiotemporal scales, as highlighted by Duan et al. (2019) and Wang et al. (2019). Various methods of coarse-graining have been employed, tailored to the specific characteristics of the systems under consideration and the desired degree of precision. Generally, CG frameworks can be classified into top-down and bottom-up methods. Top-down approaches are based on a higher-level description of the system and utilize macroscopic properties, including overall structural properties, thermodynamic quantities, as well as specific physical- and chemical intuitions about the system at hand. For example, the Martini force field maps heavy atoms to a single CG interaction site, which are parametrized to reproduce thermodynamic properties. Nevertheless, the model's intentional design omits a robust CG mapping from atomistic degrees of freedom and fails to adequately represent the underlying nature of molecular interactions in the CG representation (Jin et al. (2022)). On the other hand, bottom-up coarse-graining methods utilize microscopic information based on rigorous statistical mechanics principles like the potential of mean force (PMF) to parameterize the interaction potential in the mapped CG ensemble. Some of the most popular Bottom-Up approaches include Iterative Boltzmann inversion (Reatto et al. (1986); Reith et al. (2003)), relative entropy minimization (Chaimovich & Shell (2010); Mashayak et al.

(2015)), and force matching (Ercolessi & Adams (1994)). Although these Bottom-Up methods have been immensely useful in parameterizing CG potentials, they were classically limited to simpler pairwise potentials limiting their applicability to simpler systems. Additionally, methods such as Iterative Boltzmann Inversion and Relative Entropy have exhibited significant computational demands because of their iterative nature. Recent times have witnessed a growing inclination towards data-driven and machine-learning methodologies, which have enabled the mitigation of some of these challenges. Deep Inverse Liquid-State Theory (Moradzadeh & Aluru (2019); Jeong et al. (2022)) can be one such example in data-driven coarse-graining. This framework employs a deep neural network (DNN) to estimate the Lennard-Jones (LJ) force field parameters for particles based on a given RDF and thermodynamic state. Being fully data-driven, DeepILST provides the force field parameters in a non-iterative one-shot manner. On the other hand, Machine learning can be used for parameterizing atomic potentials resulting in powerful Neural Network Potentials (NNP). One of the popular frameworks utilizing NNP's is DeePCG, where a neural network is trained via a Force Matching method. DeePCG predicts the force vector acting on oxygen atoms using atomic neighborhood descriptors as inputs to the Neural Network. More recently, there have been intriguing advancements in coarse-graining frameworks based on the Neural Ordinary Differential Equation (Chen et al. (2018); Wang et al. (2020; 2023a)). These methods involve parameterizing neural network potentials to match target simulation output quantities by differentiating through molecular trajectories. With these advancements NNPs have offered remarkable universality and flexibility capturing a wide range of systems with accuracy comparable to high resolution atomistic or ab initio methods. However, most CG frameworks that preserve structure fail to capture dynamics accurately, and many CG models exhibit significantly faster diffusion compared to their AA counterparts (Guenza et al. (2018)). Eliminating degrees of freedom from the system alters the interplay between different dynamical processes, leading to a smoothing of the free energy landscape, which generally accelerates the dynamics (Rudzinski (2019)). Since the energy barrier hinders a system from surmounting the barrier and has the system trapped in the state near the minima (Stillinger & Weber (1982)), it can be deduced that a system shows faster dynamics with a smoothened free energy landscape. It is also discussed that for supercooled liquids, kinetics can be understood (or optimized) by quantifying the energy landscape characteristics of the system (Sasai (2003)). Another simpler perspective can be presented in terms of mean force and force distribution: the smoothened landscape and its difference from the original landscape can be seen as mean force and force fluctuation. In conventional CG frameworks, the force fluctuation is not reflected, resulting in unphysical dynamics in CG systems. While the faster dynamics could be advantageous for a specific purpose, such as fast equilibration (Kmiecik et al. (2016)), the altered dynamics of CG system is generally an undesired aspect in molecular simulation as it obfuscates true dynamical response of the system. Alternative approaches have been suggested using the Mori-Zwanzig (MZ) formulation and Generalized Langevin Equation (GLE) (Guenza et al. (2018); Markutsya et al. (2022)). However, it is important to note that using those formulations introduces non-Markovian features; memory kernels and random forces are hard to compute. In this study, we propose a novel method motivated by the observation that high-frequency (HF) potential energy surface (PES) landscape features are lost in CG modeling. The introduced potential energy function restores essential energy landscape characteristics of the fine-grained model of the all-atom model. We also show that the function can be seamlessly integrated into neural network potentials due to its orthogonality to the radial distribution function (RDF). Our approach enables the simultaneous reproduction of both the structure and dynamics of the all-atom model within the CG space, thus bridging the gap between accuracy and computational efficiency in coarse-graining.

## 2 METHOD

In this section, we outline the essential steps of our approach to optimize the structure and dynamics of our CG model, as depicted in Figure 1. We first generate the phase space trajectory of the All-Atom system using Molecular Dynamics following Hamiltonian Equations of motion,

$$m_i \ddot{x}_i = f_i \qquad (\text{for } i \in N_{atom}) \qquad (1)$$

where $m_i$, $\ddot{x}_i$, $f_i$, and $N_{atom}$ are per-atom mass, per-atom acceleration, per-atom force, and the number of atoms in a system, respectively. Once a suitable mapping operator has been chosen, the same equations can describe the trajectory of the CG system. The per-atom force is the negative derivative of total potential energy with respect to cartesian coordinates, $-\frac{\partial U_{Tot}}{\partial x_i}$. Here, the

total potential energy, $U_{Tot}$, is the sum of GNN, Lennard-Jones, and high-frequency PES energy terms: $U_{GNN}(r, \theta_{GNN})$, $U_{LJ}(r, C_{12})$, and $U_{HF}(r, A, \omega)$ terms. These functions take phase space configuration as input and return potential energy.

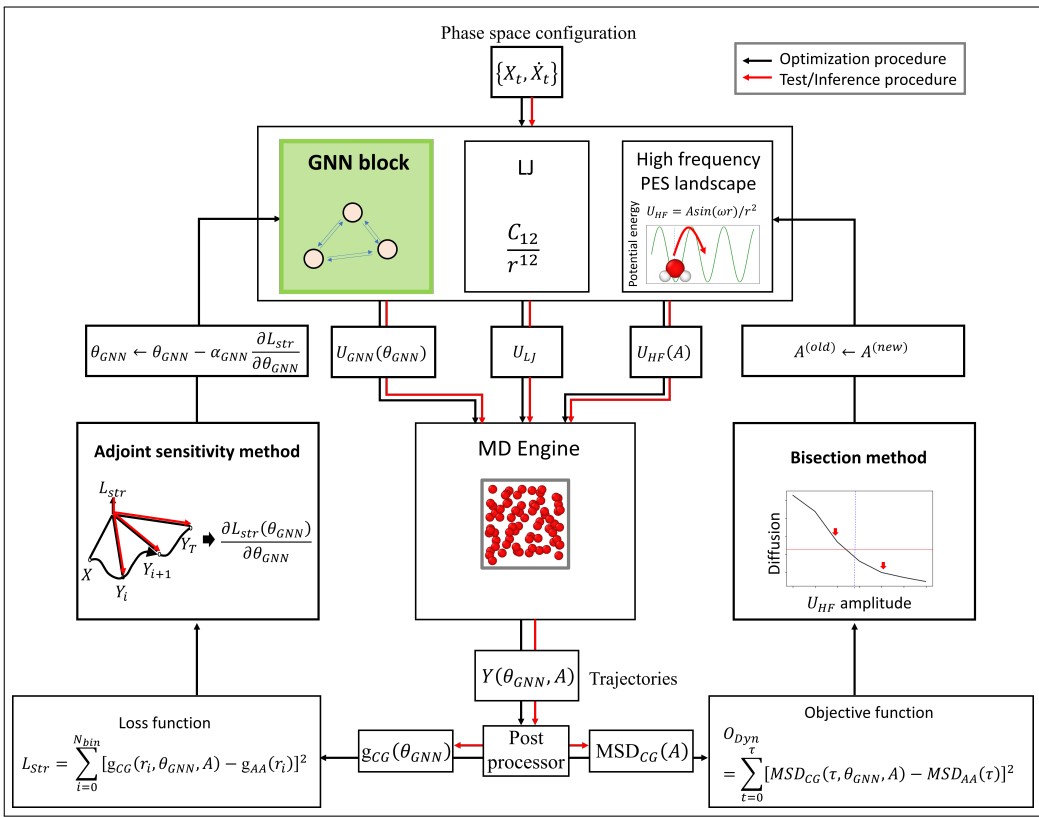

Figure 1: Workflow schematic for optimizing potential parameters of a coarse-grained model (black line) along with the inference procedure (red line). In the optimization procedure, we start with defining the potential energy function as the sum of a graph neural network potential (GNN, see Figure 2 and 3 for neural network architecture), a Lennard-Jones (LJ) potential term of which parameters are obtained from relative entropy minimization, and high-frequency potential energy surface (PES) landscape term. Using the initial phase space configuration, we compute these potential terms and the corresponding forces and carry Newtonian Dynamics to obtain the trajectory. Upon termination of the simulation, the trajectories are post-processed to extract the radial distribution function ($g_{CG}$) and the mean-squared displacement ($MSD_{CG}$). The loss functions, $L_{Str}$ and $O_{Dyn}$, are computed as the mean-squared error between the CG and target systems (all-atom model). The AA RDF and MSD can also be derived from experimentally measurable quantities, such as structure factor and diffusion coefficients. GNN parameters are optimized via Neural ODE formulation, where the adjoint sensitivity method is used to compute loss function gradient. Once RDF optimization is completed, the high-frequency PES landscape parameters, i.e., amplitude and frequency, are determined via the bisection method, iteratively (see Figure 4). Note that determining the PES landscape parameters doesn't perturb RDF, and the algorithm can always find the root since naïve CG dynamics is faster than the AA counterparts; dynamics monotonically slows down with respect to the amplitude of PES landscape term (see Figure 4 and Section 2.3).

We perform the MD simulation using a Velocity Verlet method with a time step of 1 fs as the integration scheme in an NVT (canonical) ensemble maintained at T = 300K. The temperature is maintained using the Nosé-Hoover Thermostat (Nosé (1984); Hoover (1985)). The system comprises of N atoms enclosed within a cubic box, which is subject to periodic boundary conditions. After the simulation is done, we post-process the trajectories to obtain the RDF and the mean-squared displacement.

The loss functions of RDF and MSD, $L_{Str}$ and $O_{Dyn}$, are computed as the mean squared error between the CG and target systems (AA model). The MSD is computed using the Einstein Relation, and the RDF is calculated by binning the pairwise distances between atoms. Because the histogram obtained by binning might not be smooth and non-differentiable, we apply a Gaussian smearing function to pair distance distributions, making the histogram differentiable for pair distance as previously done by (Wang et al. (2020; 2023a); Thaler & Zavadlav (2021)). GNN parameters $\theta_{GNN}$ are optimized for $L_{Str}$ via Neural ODE formulation with the adjoint sensitivity method, which enables an end-to-end trajectory differentiation to optimize GNN parameters for RDF. Compared to naïve backpropagation, the Neural ODE approach has a smaller numerical error and is memory efficient (Chen et al. (2018)). Adam optimizer (Kingma & Ba (2017)) is used to compute loss function gradient and update GNN parameters. The parameters defining the high-frequency potential energy surface (PES) landscape—specifically, the amplitude and frequency—are refined and ascertained for $O_{Dyn}$ via the bisection method. The orthogonality between high-frequency PES and RDF (see Figure 4) leads to a monotonic optimization objective amenable to such a method. We used and modified TorchMD (https://github.com/torchmd) and mdgrad (https://github.com/torchmd/mdgrad) libraries for the MD simulation procedure and neural ODE feature, respectively. Our code is available at https://sites.google.com/view/code-upload/home?authuser=3.

## 2.1 GNN STRUCTURE

Our GNN structure comprises of three operations: graph construction, node embedding, and message passing. Once the global graph containing the state of a system is constructed, it is fed into one node embedding layer and two message-passing layers. Message-passing layers are widely used in various GNN architectures for different applications (Wang et al. (2020); Park et al. (2021); Wang et al. (2023a;b); Sanchez-Gonzalez et al. (2020)), ranging from molecular scale simulation to macroscopic simulations. Next, the sum of node attributes of the final message-passing layer are computed. The final node attribute is designed to predict the per-molecule energy of each pseudo-atom.

$$U_{GNN} = \sum_{i}^{N_{atom}} h_i^{(final)} \tag{2}$$

where $h_i^{(final)}$ is $i^{th}$ node attribute in after the final message passing operation. The details of the three operations are explained below, alongside Figure 2 and 3.

The GNN block begins with graph construction. We construct a global graph data structure, denoted as $G(V, E)$, which contains the physical information of the phase space state, where $V$ and $E$ represent node and edge attributes, respectively. The node attributes contain molecule type ID ($z_i$ where $i \in N_{atom}$), and the edge attributes contain pair-wise distances ($d_{ij}$) with a cutoff distance of $r_c$ which we used 6.0 Å for CG water model.

The graph's node attributes, represented as non-continuous integers, are transformed into a continuous feature vector space using the embedding layer. The embedding parameter, $\theta_{GNN}^{Emb}$, has the size of $N_{mol\text{-}type} \times D_{Emb}$, which are the number of molecule types and the dimension of embedded feature space, respectively. In our case, we use 100 as $N_{mol\text{-}type}$ and 16 as $D_{Emb}$. It's important to note that $N_{mol\text{-}type}$ must be equal to or larger than the number of molecule types used in the target system. The output of the node embedding layer, $G(H^{(Emb)}, E)$, is the combination of updated node attributes and edge attributes, which is then fed into a series of message-passing layers.

The message-passing operation is defined as follows:

$$h_i^{new} = h_i^{old} + \sum_{j}^{N_{edge}} Message_i^{(j)} \tag{3}$$

where $h_i^{new}$ and $h_i^{old}$ refer to the node attribute of the $i^{th}$ node after and before the operation, respectively. $Message_i^{(j)}$ is $j^{th}$ row in the Message matrix. Its feature-wise (column-wise) summation for $N_{edge}$ neighbors (columns) is added to $h_i^{old}$, obtaining $h_i^{new}$. The $Message$ is generated through three $SubNet$ operations:

$$Message_i = SubNet_1(\theta_{GNN}^{sub1}, h_i^{old}) \odot SubNet_2(\theta_{GNN}^{sub1}, E_i) \tag{4}$$

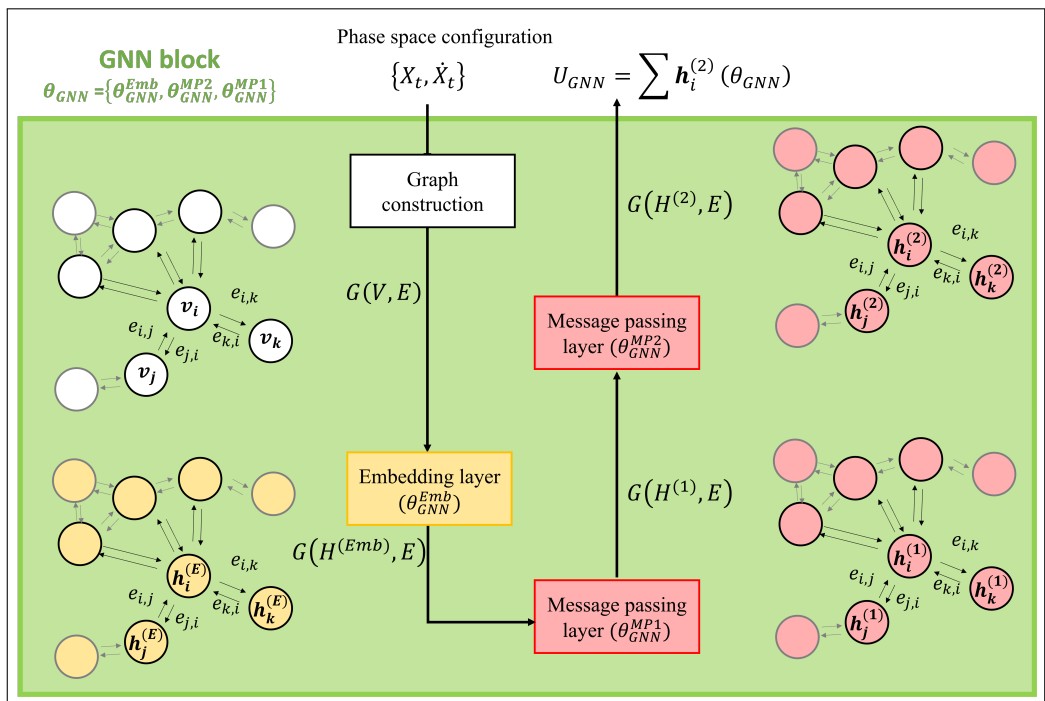

Figure 2: The architecture for the GNN block is composed of graph construction operation, embedding layer, and message passing layers. The input and output of this block are phase space configuration and GNN potential energy. The trainable parameters of this block are symbolized as $\theta_{GNN}$ which is the concatenation of embedding layer parameters ($\theta_{GNN}^{Embed}$) and message passing layer parameters ($\theta_{GNN}^{MP}$). The operation begins with the construction of a graph ($G(V, E)$) from the given position ($X_t$), followed by one embedding layer and two sequential message-passing (MP) layers. The operation details are shown in Figure 3. The output of the GNN block is obtained as the sum of the node attributes after the final MP layer.

where $Message \in \mathbb{R}^{D_{message} \times N_{edge}}$. $D_{message}$ and $N_{edge}$ refer to the feature dimensionality of the message and the number of edges connected to a vertex, respectively; $h_i^{old}$ and $E_i^{old}$ indicate the node attribute of node $i$ and the attributes of neighboring edges, respectively. $\odot$ denotes edge-wise multiplication. The $SubNet$ operations are Multi-layer perceptrons (MLPs), which can be written as follows. $SubNet_1(\theta_{GNN}^{sub1}, \cdot)$, $SubNet_2(\theta_{GNN}^{sub2}, \cdot)$, and $SubNet_3(\theta_{GNN}^{sub3}, \cdot)$

$$SubNet_1(\theta_{GNN}^{sub1}, h_i) = fc(\theta_{GNN}^{sub1}, h_i) \tag{5}$$

$$SubNet_2(\theta_{GNN}^{sub2}, h_i) = fc(\theta_{GNN}^{sub2_2}, \phi(fc(\theta_{GNN}^{sub2_1}, \rho_G(h_i)))) \tag{6}$$

$$SubNet_3(\theta_{GNN}^{sub3}, h_i) = fc(\theta_{GNN}^{sub3}, h_i) \tag{7}$$

where $fc$ and $\phi$ are fully connected layer and shifted softplus activation function. Having updated node attribute, $H^{new}$, we construct the updated graph: $G^{new}(H^{new}, E)$.

## 2.2 LENNARD-JONES POTENTIAL

Incorporating LJ (Lennard-Jones) terms in the total potential energy is crucial for addressing the data imbalance problem in MD simulations, which often leads to poor neural network training at shorter pairwise distances (Wang et al. (2019); Husic et al. (2020); Thaler & Zavadlav (2021)). The probability of observing a specific range of pairwise distances is lower for shorter distances due to high repulsion. This leads to a scarcity of training data points in extremely close proximity, resulting in a poor representation of the repulsive atomic core. By including LJ terms, the repulsive behavior is correctly accounted leading to a well-regularized neural network potential. While the LJ potential can be parameterized in different ways, we follow a physically consistent route to

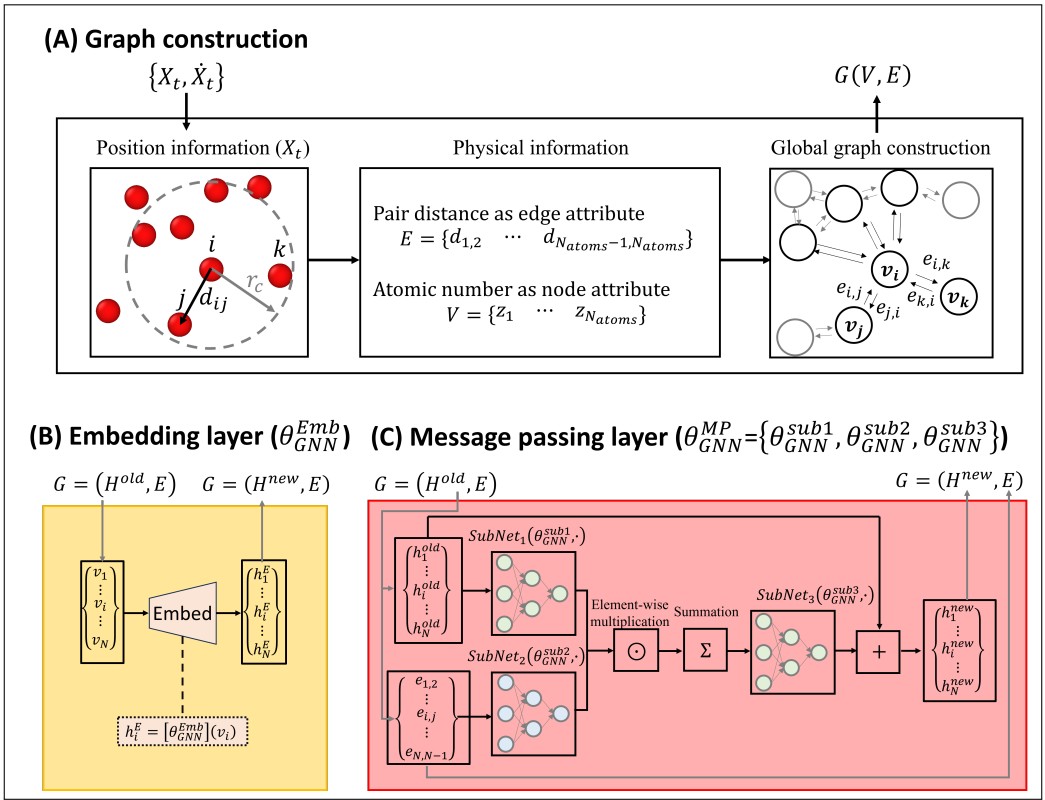

Figure 3: Three operations used in the GNN block. (A) Graph construction takes position information of oxygen atoms, $X_t$, as input and outputs a global graph, $G$, to describe the physical configuration. The graph is composed of edge attributes, $E$, and node attributes, $V$. Here, the edge attribute vector contains pairwise distance within the cutoff radius of $r_c$, and the node attribute vector has the molecule type ID. (B) The embedding layer is used to map non-continuous integer information to continuous feature vector space. (C) The message-passing layer is composed of three sub-networks ($SubNet_1$, $SubNet_2$, $SubNet_3$). The first two take node attribute vector and edge attribute vectors, respectively, and map to latent space. Then, element-wise multiplication and summations are sequentially performed to obtain the message vector. This is fed into $SubNet_3$, which maps the vector to the feature space of the node attribute. The output of $SubNet_3$ is added with the original node attribute and combined with the edge vector to be returned as the updated graph.

compute LJ parameters based on Relative Entropy Minimization. The RE framework optimizes pair potential function to minimize phase space overlap between the CG and AA ensemble, thereby giving physically meaningful LJ parameters.

## 2.3 HIGH-FREQUENCY PES LANDSCAPE

Reducing the system's degrees of freedom introduces a shift in the interplay among various dynamic processes, culminating in a smoothening of the free energy landscape. Considering the energy barrier that obstructs a system from transcending a hurdle and becoming entrapped in a proximate state adjacent to the minima (Stillinger & Weber (1982)), it is plausible to infer that a system demonstrates swifter dynamics when navigating a more smoothed free energy landscape. To correct for this behavior and recover the AA "roughness" in the PES, we introduce sinusoidal perturbations of the form,

$$U_{HF} = \frac{A sin(\omega r)}{r^2} \qquad (8)$$

where A and $\omega$ are potential parameters. $r^{-2}$ is used to ensure that the interaction strength must decrease with distance. Determining the value of omega is based on the condition that the $U_{HF}$

term is orthogonal to RDF space: RDF orthogonality. To demonstrate this, we utilize the concept of Relative Entropy or Kullback-Leibler (KL) divergence between the probability distribution of each microstate.

$$S_{rel} = \sum_i P_{AA} log \frac{P_{AA}}{P_{CG}} + \langle S_{map} \rangle_{AA} \tag{9}$$

where $P_{AA}$ and $P_{CG}$ indicate the configurational probabilities of AA and CG system, and $\langle S_{map} \rangle_{AA}$ refers to the ensemble-averaged mapping entropy in AA space, which originates due to the degeneracy caused by the mapping operator. The probability distribution is given by Boltzmann distributions of $P_{AA}$ and $P_{CG}$.

$$P(U_i) = \frac{e^{-\beta U_i}}{\sum_j e^{-\beta U_j}} \tag{10}$$

where $U_i$ refers to the potential energy of state $i$. We can rewrite equation (9), combined with equation (10):

$$S_{rel} = \beta \langle U_{CG} - U_{AA} \rangle_{AA} - \beta(F_{CG} - F_{AA}) + \langle S_{\mathrm{map}} \rangle_{AA} \tag{11}$$

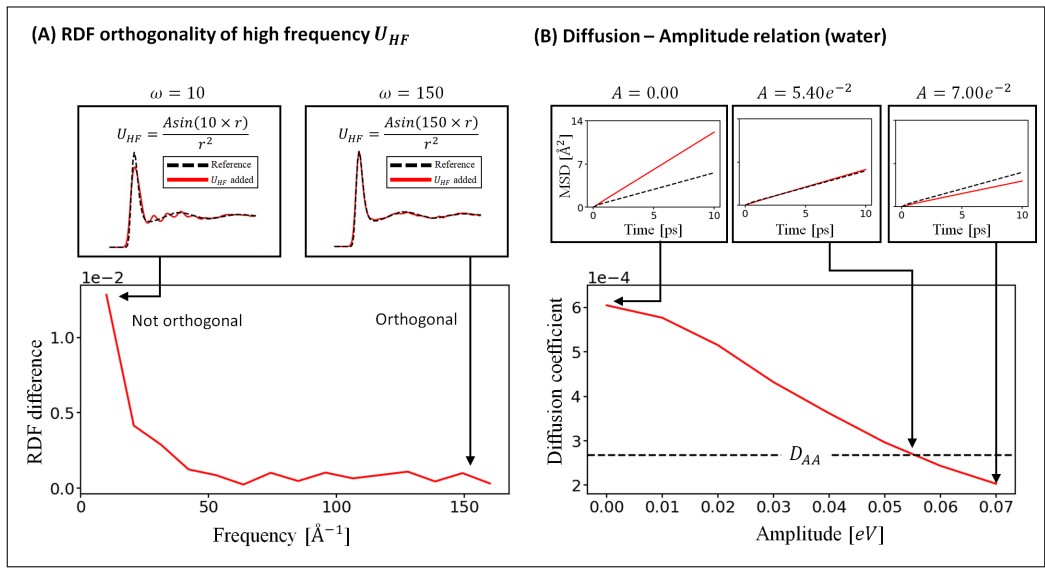

Figure 4: The relation between high-frequency PES landscape parameters and structure and dynamics of CG system. (A) The contribution of $U_{HF}(A, \omega)$ to RDF decreases as the frequency, $\omega$, increases. At high enough frequency (or wave number), for example, $150\,\text{Å}^{-1}$, this shows a negligible contribution to RDF, which we refer to as orthogonality between RDF and high-frequency PES landscape. This indicates our function can be added to other CG frameworks to reproduce dynamics accurately. (B) $U_{HF}(A, \omega)$ with high amplitude, $A$, have a molecule experiencing higher potential energy barriers, thus delaying diffusion. This monotonic relationship provides crucial information as it allows us to obtain the effective gradient for diffusion using the bisection method without the need for analytical calculation of the gradient with respect to the amplitude.

Here, $U_{CG}$, $U_{AA}$, $F_{CG}$, and $F_{AA}$ refer to the potential energies and free energies of CG and AA systems, respectively. The gradient of relative entropy with respect to potential parameters ($A$ and $\omega$) can be written as follows (details in appendix).

$$\frac{\partial S_{rel}}{\partial A} = 2\pi\rho N_{atom} \frac{sin(\omega r_c)}{\omega} \tag{12}$$

$$\frac{\partial S_{rel}}{\partial \omega} = 2\pi\rho N_{atom}(-\frac{A}{\omega^2}sin(\omega r_c) + \frac{A}{\omega}r_c cos(\omega r_c)) \tag{13}$$

It can be observed that for sufficiently large values of $\omega$, the gradients of relative entropy with respect to potential parameters approach zero. This indicates that the configurational probabilities of AA and CG systems (thus, RDFs of these systems) are maximally aligned. Through subsequent

parametric analyses, we illustrate the impact of $\omega$ on both system structure and dynamics as shown in Figure 4. As the value of $\omega$ increases, the deviation of RDF from the reference system diminishes, culminating in what can be termed an "invariance threshold" at around 50 Å$^{-1}$, beyond which RDF scarcely undergoes further change. Consequently, an optimal choice for $\omega$ emerges at 100 Å$^{-1}$.

The amplitude of $U_{HF}$ is optimized by utilizing the molecule diffusion being hindered by potential energy barriers. The hills and cleavages of $U_{HF}$ function act like potential energy barriers, and the greater the difference in their heights, the more challenging it becomes for substances to undergo diffusion. To understand the effect of hindrance and slower diffusion, we conduct a parametric study, as shown in Figure 4 (B). Initiating with the initial interval of [0 eV, 0.2 eV], we employ the bisection method to determine the optimal amplitude.

## 3 RESULT

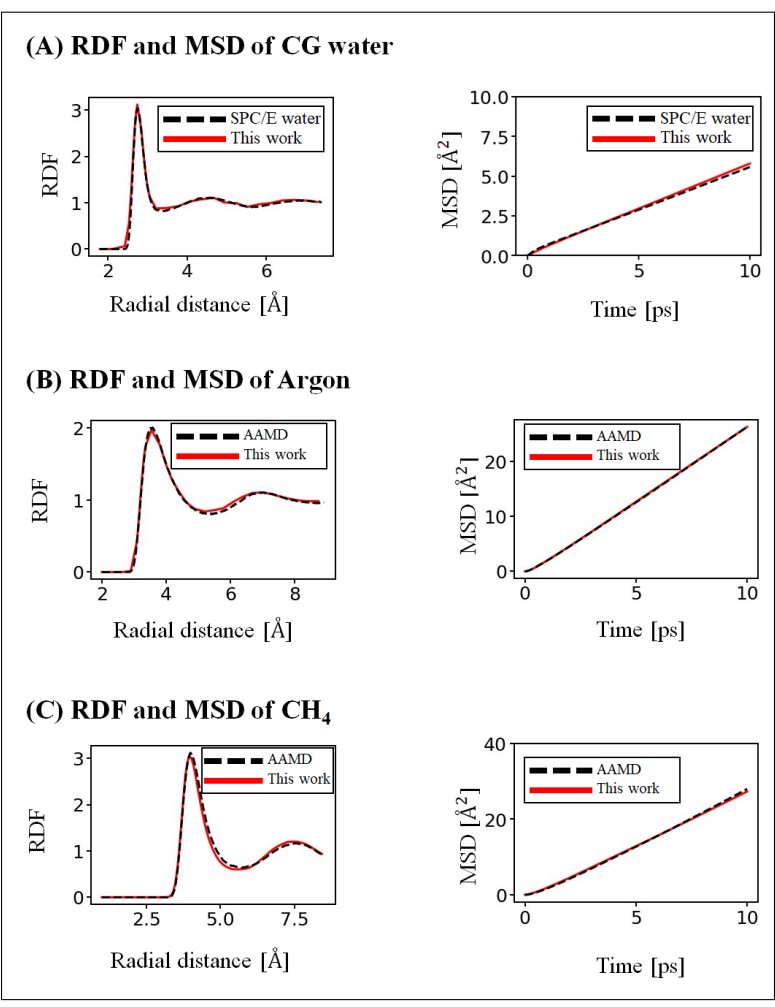

Figure 5: Comparison between all-atom RDF and MSD and their reproduction in CG simulation for various molecules: (A) SPC/E water, (B) Argon, (C) CH$_4$.

We compute the RDF and MSD from a long-time trajectory to evaluate the simulation accuracy with proper statistics (see Figure 5). We exemplify our methodology through the coarse-graining of different atomic fluids, most notably the widely employed SPC/E water model. The parameter values for this system are as follows: $\sigma_O = 3.166$ Å, $\epsilon_O = 0.650$ kJ/mol$^{-1}$, $r_{OH} = 1.000$ Å, $\angle_{HOH} = 109.47°$, $q_O = -0.8476$ $e^-$, and $q_H = 0.4238$ $e^-$ (to maintain charge neutrality). $\sigma_H$ and $\epsilon_H$ are all zero; $\sigma_{OH}$ and $\epsilon_{OH}$ are determined by Lorentz-Berthelot rules.

We compute RDF based on 100 ps of trajectory and MSD from the same trajectory with a maximum time delay of 10 ps. The results are shown in Figure 5 (A). We also test for Argon and $CH_4$ systems: $\sigma_{Ar}$ = 3.405 Å, $\epsilon_{Ar}$ = 0.238 kJ/mol$^{-1}$, $q_{Ar}$ = 0.00, $\sigma_C$ = 3.345 Å, $\epsilon_C$ = 0.066 kJ/mol$^{-1}$, $\sigma_H$ = 0.00 Å, $\epsilon_H$ = 0.00 kJ/mol$^{-1}$, $q_C^{(CH4)}$ = -0.24 $e^-$, $q_H^{(CH4)}$ = -0.06 $e^-$.

## 4 CONCLUSION

In this study, we developed a coarse-grained (CG) model for an all-atom water system and demonstrated its ability to reproduce the structure and dynamics of the all-atom model. Our approach utilized a graph neural network potential and a high-frequency potential energy surface landscape function to optimize parameters that capture both structure and dynamics.

One of the key contributions of our method is incorporating the high-frequency potential energy surface (PES) landscape. This restores the lost high-frequency characteristics in CG modeling and does not perturb RDF when the RDF-orthogonality condition is satisfied. By optimizing the parameters of the high-frequency PES, we enhanced the CG model's accuracy in capturing the all-atom system dynamics. This dual proficiency in handling both structure and dynamics is vital for various applications involving molecular-level physical, chemical, and biological processes.

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

## A  RDF ORTHOGONALITY MATH PROOF

Here, we derive from equation 11 to obtain equations 12 and 13. Gradient of $S_{rel}$ with respect to potential parameters can be simplified as follows.

$$\nabla_{A,\omega} S_{rel} = \nabla_{A,\omega} \beta \langle U_2 - U_1 \rangle_1$$

Expanding $\beta \langle U_2 - U_1 \rangle_1$

$$\beta \langle U_2 - U_1 \rangle_1 = \beta \int_0^{r_{cut}} [U_2 - U_1] g(r) dv(r)$$

$$= \beta \int_0^{r_{cut}} [U_2 - U_1] 4\pi r^2 g(r) dr$$

$$= 2\beta\pi\rho N \int_0^{r_{cut}} \frac{A\sin(\omega r)}{r^2} r^2 g(r) dr$$

$$= 2\beta\pi\rho N \times \int_0^{r_{cut}} A\sin(\omega r) g(r) dr$$

The integration can be modified as follows.

$$\int_0^{r_{cut}} \sin(\omega r) g(r) dr = \left. -g(r)\left(\frac{1}{\omega}\cos(\omega r)\right)\right|_0^{r_{cut}} + \int_0^{r_{cut}} g'(r)\left(\frac{1}{\omega}\cos(\omega r)\right) dr$$

The second term on the right-hand side can be expanded as follows.

$$\int_0^{r_{cut}} g'(r)\frac{1}{\omega}\cos(\omega r) dr = \left. g'(r)\left(\frac{1}{\omega^2}\sin(\omega r)\right)\right|_0^{r_{cut}} - \int_0^{r_{cut}} g''(r)\left(\frac{1}{\omega^2}\sin(\omega r)\right) dr$$

Expanding the second term can be

$$-\int_0^{r_{cut}} g''(r)\left(\frac{1}{B^2}\sin(Br)\right) dr = \left. g''(r)\left(\frac{1}{B^3}\cos(Br)\right)\right|_0^{r_{cut}} - \int_0^{r_{cut}} g'''(r)\left(\frac{1}{B^3}\cos(Br)\right) dr$$

Repeating this expansion can be summarized as follows.

$$\int_0^{r_{cut}} \sin(Br) g(r) dr \quad = \left. -g(r)\left(\frac{1}{B}\cos(Br)\right)\right|_0^{r_{cut}} + \left. g'(r)\left(\frac{1}{\omega^2}\sin(\omega r)\right)\right|_0^{r_{cut}}$$

$$- \int_0^{r_{cut}} g''(r)\left(\frac{1}{\omega^2}\sin(\omega r)\right) dr + \left. g''(r)\left(\frac{1}{B^3}\cos(Br)\right)\right|_0^{r_{cut}}$$

$$- \left. g'''(r)\left(\frac{1}{B^4}\sin(Br)\right)\right|_0^{r_{cut}} + \cdots$$

The right-hand side is composed $g^{(i)}$ with different $i$, which can be written as follows

$$g(r) = \exp(-\beta w(r))$$

$$g'(r) = -\beta \frac{dw}{dr} g(r)$$

$$g''(r) = -\beta \frac{dw}{dr} g' - \beta \frac{d^2 w}{dr^2} g$$

At $r = 0$, considering that $g(0)$ is zero, $g^{(i)}$ for $i > 0$ is equal to zero. At $r = r_{\text{cut}}$, $g(r_{\text{cut}}) = 1$ and $g^{(i)}$ for $i > 0$ is zero since there is no more correlation between atom pairs; thus, no more correlation function change beyond $r_{cut}$. Thus, $\nabla_{A,\omega} S_{rel}$ can be simplified as follows.

$$\nabla_{A,\omega} S_{rel} = \nabla_{A,\omega} \text{ Const } \times A \frac{g\left(r_{cut}\right)}{\omega} \cos\left(\omega r_{cut}\right)$$

Expanding for each potential parameter, we obtain the following equations, which are equations 12 and 13 , respectively.

$$\nabla_A S_{rel} = \text{ Const } \times \frac{1}{\omega} \sin\left(\omega r_{cut}\right)$$
$$\nabla_\omega S_{rel} = \text{ Const } \times \left(-\frac{A}{\omega^2} \sin\left(\omega r_{cut}\right) + \frac{A}{\omega} r_{cut} \cos\left(\omega r_{cut}\right)\right)$$

