# OpenReview forum: "ACHIEVING DYNAMIC ACCURACY IN MACHINE-LEARNED CG POTENTIALS BY MODULATING POTENTIAL ENERGY LANDSCAPE"
_ICLR.cc/2024/Conference — ICLR 2024 Conference Withdrawn Submission_

### Official Review · Reviewer_fWpi · 2023-10-22

**Soundness:** 3 good
**Presentation:** 4 excellent
**Contribution:** 2 fair
**Rating:** 5
**Confidence:** 4

**Summary:**

This paper presents a coarse-graining approach using GNN potentials combined with high-frequency potential energy surface (PES) modulation. The goal is to improve accuracy of structure and dynamics. The GNN captures complex atomic interactions and structural properties. The PES modulation recovers high-frequency landscape features lost in GNN to enable accurate dynamics, in particular, to address the problem of fast diffusion in CG system. The PES term is designed to be approximately orthogonal to radial distribution function, such that the addition of this term can improve the dynamics without degradation of structural properties. The authors demonstrate the approach by coarse-graining simple bulk systems water, CH4 and argon, and showing accurate reproduction of reference radial distribution functions and mean-squared displacements.

**Strengths:**

Originality: A novel potential energy surface modulation applied for coarse-graining molecular systems, and its theoretical results on the orthogonality with RDF

Quality: Technically sound with detailed derivations. Results clearly demonstrate accurate structure and dynamics. Well-conceived and implemented.

Clarity: Well-written and easy to follow. Logical flow with supporting diagrams and equations.

Significance: Potential impact for simulating complex biomolecular and material systems, and inspiring steps on the key challenges of structure and dynamics accuracy.

**Weaknesses:**

- The validation is limited to only simple systems. Testing on more complex systems would better demonstrate utility and scalability. Specific examples should be benchmarked.
- Transferability remains unclear. Tuning the PES amplitude for one system might not translate well to other state points or molecular species.
- The authors only train and validate on RDF and MSD, however, there are other metrics such as angular density function (ADF) that are important and not considered or tested in this work
- The mathematical derivation of RDF orthogonality lacks rigor in parts. The discarded non-zero term in the appendix needs revisiting. In the 1st equation of page 13 (in appendix), the authors basically throw away the 2nd term in the 3rd equation of this appendix. However, this is not correct, as the 2nd term would not be zero and the authors can verify it numerically. The argument using the expansion is not correct. But what remains effective is the asymptotic behavior, which the authors can expand until g'', and then argue that \int g''(r) sin(wr)/w^2 dr is of order O(1/w^2). Then the result is A g cos(wr)/w + O(1/w^2). And the conclusion when w->inf stays unchanged.
The last equation of appendix has a small error, the authors need to double check the math

**Questions:**

Some minor comments:
- Figure 1 looks like the training of GNN and HF term are parallel, but there is actually an order. This should be envisioned in the workflow
- In figure 2, the four graphs on the left and right with circles and arrows are redundant and taking extra space, since there is a graph in figure 3A showing it
- Eq.6 is confusing, the subnet2 should take edge as input and the rho_G is not explained.
- In the definition of U_HF (eq.8) the authors need to justify why they choose the form using sin() and 1/r^2 instead of other functions and powers. And since this term already diverges at r->0, why they still need the LJ term.
- The U_AA in Eq.11 actually means the CG potential + the HF potential in the context, but not the true AA potential. Need to change a notation or clarify
- The authors should note that the orthogonality is approximate/asymptotic.
- The authors should mention if there any other papers addressing the same problem, and if so, benchmark with them
- In appendix, there is a mixed use of w and B.
- On the latex format: the "word" in the equation (such as GNN, atom, final, new, old etc.) should use \mathrm instead of italic. And the index (e.g. i, j, k, t) should remain italic. Please correct all equations and notations.

---

### Official Review · Reviewer_AGGc · 2023-10-23

**Soundness:** 2 fair
**Presentation:** 2 fair
**Contribution:** 2 fair
**Rating:** 3
**Confidence:** 3

**Summary:**

This work proposes a coarse-grained (CG) model that combines a graph neural network potential and a high-frequency potential energy surface landscape function to optimize parameters that capture both structure and dynamics. Empirically, they apply their method to reproduce the RDF and MSD in CG simulation for three molecules: 1) SPC/E water, (B) Argon, (C) CH4.

**Strengths:**

The research direction of CG molecular dynamics simulations is interesting and promising, which may serve as a useful tool to advance a wide range of physics, chemistry, and biology, etc.

**Weaknesses:**

limited experiments:  As an algorithmic work that proposes a novel model architecture for CG simulation, the empirical study of it is very limited. First, whether the three systems are representative in the area of CG simulation?  Second, what is AAMD? it seems to be the only baseline, but l can not find any reference for it throughout the paper. Third, besides the simulation result, l didn't see any results about testing the generalization ability of the model, which is important for a machine learning-based potential model.

Lack of novelty in model architecture: all three operations: graph construction, node embedding, and message passing in the GNN module are conventional and lack sufficient innovation.

**Questions:**

My questions are all listed in the Weaknesses.

---

### Official Review · Reviewer_f7Ks · 2023-10-26

**Soundness:** 2 fair
**Presentation:** 1 poor
**Contribution:** 2 fair
**Rating:** 1
**Confidence:** 5

**Summary:**

In this paper, the authors propose a coarse-grained (CG) model that combines a graph neural network (GNN) potential with a high-frequency potential energy surface landscape function to reproduce the structure and dynamics of all-atom (AA) systems.

**Strengths:**

The direction explored in this paper, combining graph neural networks with coarse-grained modeling for reproducing all-atom system structures and dynamics, is indeed interesting and meaningful.

**Weaknesses:**

* Lack of novelty: The proposed approach of combining a GNN potential with a high-frequency potential energy surface landscape function is not sufficiently novel. The authors do not provide a clear explanation of how their approach differs from these existing methods and why it is expected to perform better. Moreover, the use of GNNs for modeling atomic systems has been explored in prior literature, and the authors do not demonstrate any substantial improvements over these existing methods.
* Poor writing quality: The paper suffers from poor writing quality, which makes it difficult to understand the authors' contributions and methodology. The text is often unclear, with convoluted sentence structures, and the explanations of the methods are not presented in a coherent manner. The authors should invest more effort in improving the clarity of the manuscript by providing a better organization of the content, more concise explanations of the methodology, and clearer illustrations of the concepts. For example, the introduction of the paper should be separated into several paragraphs to better present the novelty and contributions of the study. This would allow for a clearer and more organized presentation of the research background, problem statement, and the proposed solution. In the revised introduction, the authors should emphasize the novelty of their approach in combining graph neural networks with coarse-grained modeling and highlight how it addresses the limitations of existing methods. Furthermore, the authors should clearly outline the main contributions of their work, such as the specific improvements in reproducing all-atom system structures and dynamics, the demonstration of the methodology's effectiveness on various systems, and the potential impact on the broader scientific community. By restructuring the introduction in this manner, the authors can better communicate the significance of their research and its implications for the field of coarse-grained modeling.
* Lack of related references: The authors do not sufficiently cite and discuss related work, especially recent publications in the field of coarse-grained modeling and machine learning potentials for atomic systems. This makes it difficult to understand the context of the proposed approach and its significance compared to existing methods. Moreover, the lack of proper referencing indicates that the authors might not be fully aware of the current state-of-the-art in this area, which raises concerns about the novelty and relevance of their work.
* Lack of results: The experimental results presented in the paper are rather poor and insufficient to support the claims made by the authors.

**Questions:**

The related issues have been elaborated in the "weaknesses" section.

---

### Official Review · Reviewer_nqZm · 2023-11-01

**Soundness:** 3 good
**Presentation:** 3 good
**Contribution:** 2 fair
**Rating:** 3
**Confidence:** 4

**Summary:**

This paper presents a coarse-grained modeling approach aimed at matching not only thermodynamic observables but also dynamic ones. The authors employed differentiable simulation technique to train a graph neural network force field to reproduce the target radial distribution function. the authors incorporate a high-frequency energy term to perturb the coarse-grained potential energy surface. This perturbation is designed to emulate the ruggedness in all-atom energy surfaces and effectively reproduce diffusivity. Application of this method to coarse-grained simulations of simple molecules (SPC/E water, argon, and methane) demonstrates the ability to replicate both static (RDF) and dynamics (MSD) observables obtained from all-atom simulations.

**Strengths:**

The approach combines strong inductive biases about the nature of interatomic interactions in CG potentials, in particular the high frequency term.
The empirical performance on the selected systems is good

**Weaknesses:**

•	The Result Section lacks quantitative or qualitative analysis of the simulations, apart from Figure 5. At least, the differences between the RDF and MSD results obtained in this work (with and without high frequency potential term) and all-atom MD simulations should be quantified.
•	How do the trained potential and high-frequency potential terms generalize to simulations under various conditions, such as different temperatures and system sizes? It is hard to appreciate the significance of this work when the potential is trained and tested on the same simulation data.
- The test systems are too similar, molecular liquids with a single have atom and CG hydrogens.

**Questions:**

•	In page 2, the reference for DeePCG is missing.
•	In Section 2.2 and Figure 1, it is not clear whether both the Lennard-Jones attractive (r–6) and repulsive (r–12) terms are utilized in the proposed method or only the repulsive term is used.
•	It would be worth discussing why adding a high frequency potential term is advantageous compared to achieving similar diffusivity modulation by using the Langevin thermostat with different friction coefficient, instead of using the Nosé-Hoover thermostat.

---

### Meta-Review · Area_Chair_ZUpJ · 2023-12-13

**Metareview:**

The reviews for this paper were unanimous in their recommendation to not accept this paper, and the authors submitted no rebuttal. Therefore the AC recommends to reject this paper. I would like to thank the reviewers for their reviews. I hope that the reviews are helpful for the authors to improve their manuscript.

**Justification For Why Not Higher Score:**

Unanimous vote for rejecting this paper by reviewers, and no rebuttal submitted.

**Justification For Why Not Lower Score:**

N/A

---

### Decision · Program_Chairs · 2024-01-16

Reject